# CROSS-ITERATION BATCH NORMALIZATION

## ABSTRACT

A well-known issue of Batch Normalization is its significantly reduced effectiveness in the case of small mini-batch sizes. When a mini-batch contains few examples, the statistics upon which the normalization is defined cannot be reliably estimated from it during a training iteration. To address this problem, we present Cross-Iteration Batch Normalization (CBN), in which examples from multiple recent iterations are jointly utilized to enhance estimation quality. A challenge of computing statistics over multiple iterations is that the network activations from different iterations are not comparable to each other due to changes in network weights. We thus compensate for the network weight changes via a proposed technique based on Taylor polynomials, so that the statistics can be accurately estimated and batch normalization can be effectively applied. On object detection and image classification with small mini-batch sizes, CBN is found to outperform the original batch normalization and a direct calculation of statistics over previous iterations without the proposed compensation technique.

## 1 INTRODUCTION

Batch Normalization (BN) (Ioffe & Szegedy, 2015) has played a significant role in the success of deep neural networks. It was introduced to address the issue of internal covariate shift, where the distribution of network activations changes during training iterations due to the updates of network parameters. This shift is commonly believed to be disruptive to network training, and BN alleviates this problem through normalization of the network activations by their mean and variance, computed over the examples within the mini-batch at each iteration. With this normalization, network training can be performed at much higher learning rates and with less sensitivity to weight initialization.

In BN, it is assumed that the distribution statistics for the examples within each mini-batch reflect the statistics over the full training set. While this assumption is generally valid for large batch sizes, it breaks down in the *small batch size regime* (Peng et al., 2018; Wu & He, 2018; Ioffe, 2017), where noisy statistics computed from small sets of examples can lead to a dramatic drop in performance. This problem hinders the application of BN to memory-consuming tasks such as object detection (Ren et al., 2015; Dai et al., 2017), semantic segmentation (Long et al., 2015; Chen et al., 2017) and action recognition (Wang et al., 2018b), where batch sizes are limited due to memory constraints.

Towards improving estimation of statistics in the small batch size regime, alternative normalizers have been proposed. Several of them, including Layer Normalization (LN) (Ba et al., 2016), Instance Normalization (IN) (Ulyanov et al., 2016), and Group Normalization (GN) (Wu & He, 2018), compute the mean and variance over the channel dimension, independent of batch size. Different channel-wise normalization techniques, however, tend to be suitable for different tasks, depending on the set of channels involved. On the other hand, synchronized BN (SyncBN) (Peng et al., 2018) yields consistent improvements by processing larger batch sizes across multiple GPUs. These gains in performance come at the cost of additional overhead needed for synchronization across the devices.

A seldom explored direction for estimating better statistics is to compute them over the examples from multiple recent training iterations, instead of from only the current iteration as done in previous techniques. This can substantially enlarge the pool of data from which the mean and variance are obtained. However, there exists an obvious drawback to this approach, in that the activation values from different iterations are not comparable to each other due to the changes in network weights. As shown in Figure 1, directly calculating the statistics over multiple iterations, which we refer to as Naive CBN, results in lower accuracy.

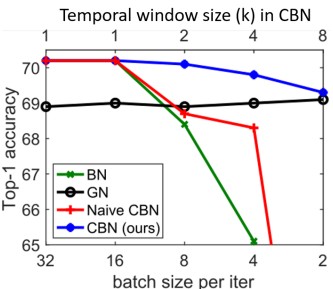

Figure 1: **Top-1 classification accuracy *vs*. batch sizes per iteration.** The base model is a ResNet-18 (He et al., 2016) trained on ImageNet (Russakovsky et al., 2015). BN (Ioffe & Szegedy, 2015)'s accuracy drops rapidly when the batch size is reduced. GN (Wu & He, 2018) exhibits stable performance but underperforms BN on adequate batch sizes. CBN compensates for the reduced batch size per GPU by exploiting approximated statistics from recent iterations (Temporal window size denotes how many recent iters are utilized for statistics computation). CBN shows relatively stable performance over different batch sizes. Naive CBN does not work well, which directly calculates statistics from recent iterations without compensation.

In this paper, we present a method that compensates for the network weight changes among iterations, so that examples from preceding iterations can be effectively used to improve batch normalization. Our method, called Cross-Iteration Batch Normalization (CBN), is motivated by the observation that network weights change gradually, instead of abruptly, between consecutive training iterations, thanks to the iterative nature of Stochastic Gradient Descent (SGD). As a result, the mean and variance of examples from recent iterations can be well approximated for the current network weights via a low-order Taylor polynomial, defined on gradients of the statistics with respect to the network weights. The compensated means and variances from multiple recent iterations are averaged with those of the current iteration to produce better estimates of the statistics.

In the small batch size regime, CBN leads to appreciable performance improvements over the original BN, as exhibited in Figure 1. The superiority of our proposed approach is further demonstrated through more extensive experiments on ImageNet classification and object detection on COCO. These gains are obtained with negligible overhead, as the statistics from previous iterations have already been computed and Taylor polynomials are simple to evaluate. With this work, it is shown that cues for batch normalization can successfully be extracted along the time dimension, opening a new direction for investigation.

## 2 RELATED WORK

The importance of normalization in training neural networks has been recognized for decades (LeCun et al., 1998). In general, normalization can be performed on three components: input data, hidden activations, and network parameters. Among them, input data normalization is used most commonly because of its simplicity and effectiveness (Sola & Sevilla, 1997; LeCun et al., 1998).

After the introduction of Batch Normalization (Ioffe & Szegedy, 2015), the normalization of activations has become nearly as prevalent. By normalizing hidden activations by their statistics within each mini-batch, BN effectively alleviates the vanishing gradient problem and significantly speeds up the training of deep networks. To mitigate the mini-batch size dependency of BN, a number of variants have been proposed, including Layer Normalization (LN) (Ba et al., 2016), Instance Normalization (IN) (Ulyanov et al., 2016), Group Normalization (GN) (Wu & He, 2018), and Batch Instance Normalization (BIN) (Nam & Kim, 2018). The motivation of LN is to explore more suitable statistics for sequential models, while IN performs normalization in a manner similar to BN but with statistics only for each instance. GN achieves a balance between IN and LN, by dividing features into multiple groups along the channel dimension and computing the mean and variance within each group for normalization. BIN introduces a learnable method for automatically switching between normalizing and maintaining style information, enjoying the advantages of both BN and IN on style transfer tasks. Cross-GPU Batch Normalization (CGBN or SyncBN) (Peng et al., 2018) extends BN across multiple GPUs for the purpose of increasing the effective batch size. Though providing higher accuracy, it introduces synchronization overhead to the training process. Kalman Normalization (KN) (Wang et al., 2018a) presents a Kalman filtering procedure for estimating the statistics for a network layer from the layer's observed statistics and the computed statistics of previous layers.

Batch Renormalization (BRN) (Ioffe, 2017) is the first attempt to utilize the statistics of recent iterations for normalization. It does not compensate for the statistics from recent iterations, but rather it down-weights the importance of statistics from distant iterations. This down-weighting heuristic, however, does not make the resulting statistics "correct", as the statistics from recent iterations are not of the current network weights. BRN can be deemed as a special version of our Naive CBN baseline (without Taylor polynomial approximation), where distant iterations are down-weighted.

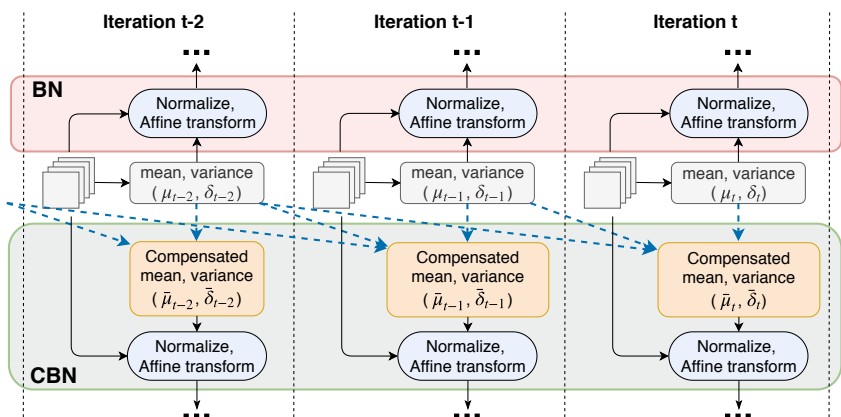

Figure 2: Illustration of BN and the proposed Cross-Iteration Batch Normalization (CBN).

Recent work have also investigated the normalization of network parameters. In Weight Normalization (WN) (Salimans & Kingma, 2016), the optimization of network weights is improved through a reparameterization of weight vectors into their length and direction. Weight Standardization (WS) (Qiao et al., 2019) instead reparameterizes weights based on their first and second moments for the purpose of smoothing the loss landscape of the optimization problem. To combine the advantages of multiple normalization techniques, Switchable Normalization (SN) (Luo et al., 2018) and Sparse Switchable Normalization (SSN) (Shao et al., 2019) make use of differentiable learning to switch among different normalization methods.

The proposed CBN takes an activation normalization approach that aims to mitigate the mini-batch dependency of BN. Different from existing techniques, it provides a way to effectively aggregate statistics across multiple training iterations.

## 3 METHOD

### 3.1 REVISITING BATCH NORMALIZATION

The original batch normalization (BN) (Ioffe & Szegedy, 2015) whitens the activations of each layer by the statistics computed within a mini-batch. Denote $\theta_t$ and $x_{t,i}(\theta_t)$ as the network weights and the feature response of a certain layer for the $i$-th example in the $t$-th mini-batch. With these values, BN conducts the following normalization:

$$\hat{x}_{t,i}(\theta_t) = \frac{x_{t,i}(\theta_t) - \mu_t(\theta_t)}{\sqrt{\sigma_t(\theta_t)^2 + \varepsilon}}, \tag{1}$$

where $\hat{x}_{t,i}(\theta_t)$ is the whitened activation with zero mean and unit variance, $\varepsilon$ is a small constant added for numerical stability, and $\mu_t(\theta_t)$ and $\sigma_t(\theta_t)$ are the mean and variance computed for all the examples from the current mini-batch, i.e.,

$$\mu_t(\theta_t) = \frac{1}{m} \sum_{i=1}^{m} x_{t,i}(\theta_t), \tag{2}$$

$$\sigma_t(\theta_t) = \sqrt{\frac{1}{m} \sum_{i=1}^{m} (x_{t,i}(\theta_t) - \mu_t(\theta_t))^2} = \sqrt{\nu_t(\theta_t) - \mu_t(\theta_t)^2}, \tag{3}$$

where $\nu_t(\theta_t) = \frac{1}{m} \sum_{i=1}^{m} x_{t,i}(\theta_t)^2$, and $m$ denotes the number of examples in the current mini-batch. The whitened activation $\hat{x}_{t,i}(\theta_t)$ further undergoes a linear transform with learnable weights, to increase its expressive power:

$$y_{t,i}(\theta_t) = \gamma \hat{x}_{t,i}(\theta_t) + \beta, \tag{4}$$

where $\gamma$ and $\beta$ are the learnable parameters (initialized to $\gamma = 1$ and $\beta = 0$ in this work).

When the batch size $m$ is small, the statistics $\mu_t(\theta_t)$ and $\sigma_t(\theta_t)$ become noisy estimates of the training set statistics, thus degrading the effects of batch normalization. In the ImageNet classification task

for which the BN module was originally designed, a batch size of 32 is typical. However, for other tasks requiring larger models and/or higher image resolution, such as object detection, semantic segmentation and video recognition, the typical batch size may be as small as 1 or 2 due to GPU memory limitations. The original BN becomes considerably less effective in such cases.

## 3.2 LEVERAGING STATISTICS FROM PREVIOUS ITERATIONS

To address the issue of BN with small mini-batches, a naive approach is to compute the mean and variance over the current and previous iterations. However, the statistics $\mu_{t-\tau}(\theta_{t-\tau})$ and $v_{t-\tau}(\theta_{t-\tau})$ of the $(t-\tau)$-th iteration are computed under the network weights $\theta_{t-\tau}$, making them obsolete for the current iteration. As a consequence, directly aggregating statistics from multiple iterations produces inaccurate estimates of the mean and variance, leading to significantly worse performance.

We observe that the network weights change smoothly between consecutive iterations, due to the nature of gradient-based training. This allows us to approximate $\mu_{t-\tau}(\theta_t)$ and $v_{t-\tau}(\theta_t)$ from the readily available $\mu_{t-\tau}(\theta_{t-\tau})$ and $v_{t-\tau}(\theta_{t-\tau})$ via a Taylor polynomial, i.e.,

$$\mu_{t-\tau}(\theta_t) = \mu_{t-\tau}(\theta_{t-\tau}) + \frac{\partial \mu_{t-\tau}(\theta_{t-\tau})}{\partial \theta_{t-\tau}}(\theta_t - \theta_{t-\tau}) + \mathbf{O}(\|\theta_t - \theta_{t-\tau}\|^2), \tag{5}$$

$$v_{t-\tau}(\theta_t) = v_{t-\tau}(\theta_{t-\tau}) + \frac{\partial v_{t-\tau}(\theta_{t-\tau})}{\partial \theta_{t-\tau}}(\theta_t - \theta_{t-\tau}) + \mathbf{O}(\|\theta_t - \theta_{t-\tau}\|^2), \tag{6}$$

where $\partial \mu_{t-\tau}(\theta_{t-\tau})/\partial \theta_{t-\tau}$ and $\partial v_{t-\tau}(\theta_{t-\tau})/\partial \theta_{t-\tau}$ are gradients of the statistics with respect to the network weights, and $\mathbf{O}(\|\theta_t - \theta_{t-\tau}\|^2)$ denotes higher-order terms of the Taylor polynomial, which can be omitted since the first-order term dominates when $(\theta_t - \theta_{t-\tau})$ is small.

In Eq. (5) and Eq. (6), the gradients $\partial \mu_{t-\tau}(\theta_{t-\tau})/\partial \theta_{t-\tau}$ and $\partial v_{t-\tau}(\theta_{t-\tau})/\partial \theta_{t-\tau}$ cannot be precisely determined at a negligible cost because the statistics $\mu_{t-\tau}^l(\theta_{t-\tau})$ and $v_{t-\tau}^l(\theta_{t-\tau})$ for a node at the $l$-th network layer depend on all the network weights prior to the $l$-th layer, i.e., $\partial \mu_{t-\tau}^l(\theta_{t-\tau})/\partial \theta_{t-\tau}^r \neq 0$ and $\partial v_{t-\tau}^l(\theta_{t-\tau})/\partial \theta_{t-\tau}^r \neq 0$ for $r \leq l$, where $\theta_{t-\tau}^r$ denotes the network weights at the $r$-th layer. Only when $r = l$ can these gradients be derived in closed form efficiently.

Empirically, we find that as the layer index $r$ decreases ($r \leq l$), the partial gradients $\frac{\partial \mu_t^l(\theta_t)}{\theta_t^r}$ and $\frac{\partial v_t^l(\theta_t)}{\theta_t^r}$ rapidly diminish. These reduced effects of network weight changes at earlier layers on the activation distributions in later layers may perhaps be explained by the reduced internal covariate shift of BN. Motivated by this phenomenon, which is studied in Appendix C, we propose to truncate these partial gradients at layer $l$.

Thus, we further approximate Eq. (5) and Eq. (6) by

$$\mu_{t-\tau}^l(\theta_t) \approx \mu_{t-\tau}^l(\theta_{t-\tau}) + \frac{\partial \mu_{t-\tau}^l(\theta_{t-\tau})}{\partial \theta_{t-\tau}^l}(\theta_t^l - \theta_{t-\tau}^l), \tag{7}$$

$$v_{t-\tau}^l(\theta_t) \approx v_{t-\tau}^l(\theta_{t-\tau}) + \frac{\partial v_{t-\tau}^l(\theta_{t-\tau})}{\partial \theta_{t-\tau}^l}(\theta_t^l - \theta_{t-\tau}^l). \tag{8}$$

A naive implementation of $\partial \mu_{t-\tau}^l(\theta_{t-\tau})/\partial \theta_{t-\tau}^l$ and $\partial v_{t-\tau}^l(\theta_{t-\tau})/\partial \theta_{t-\tau}^l$ involves computational overhead of $O(C^l \times C^l \times C^{l-1} \times K)$, where $C^l$ and $C^{l-1}$ denote the channel dimension of the $l$-th layer and the $(l-1)$-th layer, respectively, and $K$ denotes the kernel size of $\theta_{t-\tau}^l$. Here we find that the operation can be implemented efficiently in $O(C^l \times C^{l-1} \times K)$, thanks to the averaging over feature responses of $\mu$ and $v$. See Appendix B for the details.

## 3.3 CROSS-ITERATION BATCH NORMALIZATION

After compensating for network weight changes, we aggregate the statistics of the $k-1$ most recent iterations with those of the current iteration $t$ to obtain the statistics used in CBN:

$$\bar{\mu}_{t,k}^l(\theta_t) = \frac{1}{k}\sum_{\tau=0}^{k-1}\mu_{t-\tau}^l(\theta_t), \tag{9}$$

$$\bar{v}_{t,k}^l(\theta_t) = \frac{1}{k}\sum_{\tau=0}^{k-1}\max\left[v_{t-\tau}^l(\theta_t), \mu_{t-\tau}^l(\theta_t)^2\right], \tag{10}$$

|        | batch size per iter | #examples for statistics | Norm axis |
|--------|---------------------|--------------------------|-----------|
| IN     | #bs/GPU * #GPU      | 1                        | (spatial) |
| LN     | #bs/GPU * #GPU      | 1                        | (channel, spatial) |
| GN     | #bs/GPU * #GPU      | 1                        | (channel group, spatial) |
| BN     | #bs/GPU * #GPU      | #bs/GPU                  | (batch, spatial) |
| syncBN | #bs/GPU * #GPU      | #bs/GPU * #GPU           | (batch, spatial, GPU) |
| CBN    | #bs/GPU * #GPU      | #bs/GPU * **temporal window** | (batch, spatial, **iteration**) |

Table 1: Comparison of different feature normalization methods. #bs/GPU denotes batch size per GPU.

$$\bar{\sigma}_{t,k}^l(\theta_t) = \sqrt{\bar{v}_{t,k}^l(\theta_t) - \bar{\mu}_{t,k}^l(\theta_t)^2}, \tag{11}$$

where $\mu_{t-\tau}^l(\theta_t)$ and $v_{t-\tau}^l(\theta_t)$ are computed from Eq. (7) and Eq. (8). In Eq. (10), $\bar{v}_{t,k}^l(\theta_t)$ is determined from the maximum of $v_{t-\tau}^l(\theta_t)$ and $\mu_{t-\tau}^l(\theta_t)^2$ in each iteration because $v_{t-\tau}^l(\theta_t) \geq \mu_{t-\tau}^l(\theta_t)^2$ should hold for valid statistics but may be violated by Taylor polynomial approximations in Eq. (7) and Eq. (8). Finally, $\bar{\mu}_{t,k}^l(\theta_t)$ and $\bar{\sigma}_{t,k}^l(\theta_t)$ are applied to normalize the corresponding feature responses $\{x_{t,i}^l(\theta_t)\}_{i=1}^m$ at the current iteration:

$$\hat{x}_{t,i}^l(\theta_t) = \frac{x_{t,i}^l(\theta_t) - \bar{\mu}_{t,k}^l(\theta_t)}{\sqrt{\bar{\sigma}_{t,k}^l(\theta_t)^2 + \varepsilon}}. \tag{12}$$

With CBN, the effective number of examples used to compute the statistics for the current iteration is $k$ times as large as that for the original BN. In training, the loss gradients are backpropagated to the network weights and activations at the current iteration, i.e., $\theta_t^l$ and $x_{t,i}^l(\theta_t)$. Those of the previous iterations are fixed and do not receive gradients. Hence, the computation cost of CBN in back-propagation is the same as that of BN.

Replacing the BN modules in a network by CBN leads to only minor increases in computational overhead and memory footprint. For computation, the additional overhead mainly comes from computing the partial derivatives $\partial\mu_{t-\tau}(\theta_{t-\tau})/\partial\theta_{t-\tau}^l$ and $\partial v_{t-\tau}(\theta_{t-\tau})/\partial\theta_{t-\tau}^l$, which is insignificant in relation to the overhead of the whole network. For memory, the module requires access to the statistics ($\{\mu_{t-\tau}^l(\theta_{t-\tau})\}_{\tau=1}^{k-1}$ and $\{v_{t-\tau}^l(\theta_{t-\tau})\}_{\tau=1}^{k-1}$) and the gradients ($\{\partial\mu_{t-\tau}(\theta_{t-\tau})/\partial\theta_{t-\tau}^l\}_{\tau=1}^{k-1}$ and $\{\partial v_{t-\tau}(\theta_{t-\tau})/\partial\theta_{t-\tau}^l\}_{\tau=1}^{k-1}$) computed for the most recent $k-1$ iterations, which is also minor compared to the rest of the memory consumed in processing the input examples. The additional computation and memory of CBN is reported for our experiments in Table 6.

A key hyper-parameter in the proposed CBN is the temporal window size, $k$, of recent iterations used for statistics estimation. A broader window enlarges the set of examples, but the example quality becomes increasingly lower for more distant iterations, since the differences in network parameters $\theta_t$ and $\theta_{t-\tau}$ become more significant and are compensated less well using a low-order Taylor polynomial. Empirically, we found that CBN is effective with a window size up to $k = 8$ in a variety of settings and tasks. The only trick is that the window size should be kept small at the beginning of training, when the network weights change quickly. Thus, we introduce a burn-in period of length $T_{\text{burn-in}}$ for the window size, where $k = 1$ and CBN degenerates to the original BN. In our experiments, the burn-in period is set to 25 epochs on ImageNet image classification and 3 epochs on COCO object detection by default. Ablations on this parameter are presented in the Appendix.

Table 1 compares CBN with other feature normalization methods. The key difference among these approaches is the axis along which the statistics are counted and the features are normalized. The previous techniques are all designed to exploit examples from the same iteration. By contrast, CBN explores the aggregation of examples along the temporal dimension. As the data utilized by CBN lies in a direction orthogonal to that of previous methods, the proposed CBN could potentially be combined with other feature normalization approaches to further enhance statistics estimation in certain challenging applications.

## 4 EXPERIMENTS

### 4.1 IMAGE CLASSIFICATION ON IMAGENET

**Experimental settings.** ImageNet (Russakovsky et al., 2015) is a benchmark dataset for image classification, containing 1.28M training images and 50K validation images from 1000 classes. We

follow the standard setting in (He et al., 2015) to train deep networks on the training set and report the single-crop top-1 accuracy on the validation set. Our preprocessing and augmentation strategy strictly follows the GN baseline (Wu & He, 2018). We use a weight decay of 0.0001 for all weight layers, including $\gamma$ and $\beta$. We train standard ResNet-18 for 100 epochs on 4 GPUs, and decrease the learning rate by the cosine decay strategy (He et al., 2019). We use the average over 5 trials for all results. All hyper-parameters, e.g. group size of GN, are carefully tuned via cross-validation. ResNet-18 with BN is our base model. To compare with other normalization methods, we directly replace BN with IN, LN, GN, BRN, and our proposed CBN.

**Comparison of feature normalization methods.** We compare the performance of each normalization method with a normal batch size, 32, in Table 2. With sufficient data for reliable statistics, BN easily reaches the highest top-1 accuracy. Similar as the results in previous papers

|                | IN   | LN   | GN   | BN       | CBN      |
|----------------|------|------|------|----------|----------|
| Top-1 accuracy | 64.4 | 67.9 | 68.9 | **70.2** | **70.2** |

Table 2: Top-1 accuracy of **feature normalization methods** using ResNet-18 on ImageNet.

(Wu & He, 2018), IN and LN achieve significantly worse performance than BN. GN works well on image classification, but still has a small degradation of 1.2% compared with BN. Over all the methods, our CBN is the only one that is able to achieve comparable accuracy with BN, as it converges to the procedure of BN as the batch size becomes larger.

**Sensitivity to batch size.** We compare the behavior of CBN, original BN (Ioffe & Szegedy, 2015), GN (Wu & He, 2018), and BRN (Ioffe, 2017) at the same number of images per GPU on ImageNet classification. For CBN, the recent iterations are utilized so as to ensure that the number of effective examples is no fewer than 16. For BRN, the settings strictly follow the original paper. We adopt a learning rate of 0.1 for the batch size of 32, and linearly scale the learning rate by $N/32$ for a batch size of $N$.

| batch size per GPU | 32       | 16       | 8        | 4        | 2        |
|--------------------|----------|----------|----------|----------|----------|
| BN                 | **70.2** | **70.2** | 68.4     | 65.1     | 55.9     |
| GN                 | 68.9     | 69.0     | 68.9     | 69.0     | 69.1     |
| BRN                | 70.1     | 68.5     | 68.2     | 67.9     | 60.3     |
| CBN                | **70.2** | **70.2** | **70.1** | **69.8** | **69.3** |

Table 3: Top-1 accuracy of feature normalization methods with **different batch sizes** using ResNet-18 as the base model on ImageNet.

The results are shown in Table 3. For the original BN, its accuracy drops noticeably as the number of images per GPU is reduced from 32 to 2. BRN suffers a significant performance drop as well. GN maintains its accuracy by utilizing the channel dimension but not batch dimension. For CBN, its accuracy holds by exploiting the examples of recent iterations. Also, CBN outperforms GN by 0.9% on average top-1 accuracy with different batch sizes. This is reasonable, because the statistics computation of CBN introduces uncertainty caused by the stochastic batch sampling like in BN, but this uncertainty is missing in GN which results in some loss of regularization ability.

## 4.2 OBJECT DETECTION AND INSTANCE SEGMENTATION ON COCO

**Experimental settings.** COCO (Lin et al., 2014) is chosen as the benchmark for object detection and instance segmentation. Models are trained on the COCO 2017 train split with 118k images, and evaluated on the COCO 2017 validation split with 5k images. Following the standard protocol in (Lin et al., 2014), the object detection and instance segmentation accuracies are measured by the mean average precision (mAP) scores at different intersection-over-union (IoU) overlaps at the box and the mask levels, respectively.

Following (Wu & He, 2018), Faster R-CNN (Ren et al., 2015) and Mask R-CNN (He et al., 2017) with FPN (Lin et al., 2017) are chosen as the baselines for object detection and instance segmentation, respectively. For both, the 2fc box head is replaced by a 4conv1fc head for better use of the normalization mechanism (Wu & He, 2018). The backbone networks are ImageNet pretrained ResNet-50 (default) or ResNet-101, with specific normalization. Finetuning is performed on the COCO train set for 12 epochs on 4 GPUs by SGD, where each GPU processes 4 images (default). Note that the mean and variance statistics in CBN are computed within each GPU. The learning rate is initialized to be $0.02 * N/16$ for a batch size per GPU of $N$, and is decayed by a factor of 10 at the 9-th and the 11-th epochs. The weight decay and momentum parameters are set to 0.0001 and 0.9, respectively. We use the average over 5 trials for all results. All hyper-parameters, e.g. group size of GN, are carefully tuned via cross-validation.

| backbone | box head | $AP^{bbox}$ | $AP^{bbox}_{50}$ | $AP^{bbox}_{75}$ | $AP^{bbox}_{S}$ | $AP^{bbox}_{M}$ | $AP^{bbox}_{L}$ |
|---|---|---|---|---|---|---|---|
| fixed BN | - | 36.9 | 58.2 | 39.9 | 21.2 | 40.8 | 46.9 |
| fixed BN | BN | 36.3 | 57.3 | 39.2 | 20.8 | 39.7 | 47.3 |
| fixed BN | syncBN | 37.7 | 58.5 | 41.1 | 22.0 | 40.9 | 49.0 |
| fixed BN | GN | 37.8 | 59.0 | 40.8 | 22.3 | 41.2 | 48.4 |
| fixed BN | CBN | 37.7 | 59.0 | 40.7 | 22.1 | 40.9 | 48.8 |
| BN | BN | 35.5 | 56.4 | 38.7 | 19.7 | 38.8 | 47.3 |
| syncBN | syncBN | 37.9 | 58.5 | 41.1 | 21.7 | 41.5 | 49.7 |
| GN | GN | 37.8 | 59.1 | 40.9 | 22.4 | 41.2 | 49.0 |
| CBN | CBN | 37.3 | 57.7 | 39.3 | 21.9 | 40.8 | 48.2 |

Table 4: Results of **feature normalization methods** on Faster R-CNN with FPN and ResNet50 on COCO.

| Backbone | method | norm | $AP^{bbox}$ | $AP^{bbox}_{50}$ | $AP^{bbox}_{75}$ | $AP^{bbox}_{S}$ | $AP^{bbox}_{M}$ | $AP^{bbox}_{L}$ |
|---|---|---|---|---|---|---|---|---|
| | | GN | 37.8 | 59.0 | 40.8 | 22.3 | 41.2 | 48.4 |
| R50+FPN | Faster RCNN | syncBN | 37.7 | 58.5 | 41.1 | 22.0 | 40.9 | 49.0 |
| | | CBN | 37.7 | 59.0 | 40.7 | 22.1 | 40.9 | 48.8 |
| | | GN | 39.3 | 60.6 | 42.7 | 22.5 | 42.5 | 51.3 |
| R101+FPN | Faster RCNN | syncBN | 39.2 | 59.8 | 43.0 | 22.2 | 42.9 | 51.6 |
| | | CBN | 39.2 | 60.0 | 42.6 | 22.3 | 42.6 | 51.1 |
| | | | $AP^{bbox}$ | $AP^{bbox}_{50}$ | $AP^{bbox}_{75}$ | $AP^{mask}$ | $AP^{mask}_{50}$ | $AP^{mask}_{75}$ |
| | | GN | 38.6 | 59.8 | 41.9 | 35.0 | 56.7 | 37.3 |
| R50+FPN | Mask RCNN | syncBN | 38.5 | 58.9 | 42.3 | 34.7 | 56.3 | 36.8 |
| | | CBN | 38.5 | 59.2 | 42.1 | 34.6 | 56.4 | 36.6 |
| | | GN | 40.3 | 61.2 | 44.2 | 36.6 | 58.5 | 39.2 |
| R101+FPN | Mask RCNN | syncBN | 40.3 | 60.8 | 44.2 | 36.0 | 57.7 | 38.6 |
| | | CBN | 40.1 | 60.5 | 44.1 | 35.8 | 57.3 | 38.5 |

Table 5: Results with **stronger backbones** on COCO object detection and instance segmentation.

As done in (Wu & He, 2018), we experiment with two settings where the normalizers are activated only at the task-specific heads with frozen BN at the backbone (default), or the normalizers are activated at all the layers except for the early conv1 and conv2 stages in ResNet.

**Normalizers at backbone and task-specific heads.** We further study the effect of different normalizers on the backbone network and task-specific heads for object detection on COCO. CBN, original BN, syncBN, and GN are included in the comparison.

Table 4 presents the results. When BN is frozen in the backbone and no normalizer is applied at the head, the $AP^{bbox}$ score is 36.9%. When the original BN is applied at the head only and at both the backbone and the head, the accuracy drops to 36.3% and 35.5%, respectively. For CBN, the accuracy is 37.7% and 37.3% at these two settings, respectively. Without any synchronization across GPUs, CBN can achieve comparable performance with syncBN and GN, showing the superiority of the proposed approach. Unfortunately, due to the accumulation of approximation error, CBN observes a 0.4% decrease in $AP^{bbox}$ when replacing frozen BN with CBN in the backbone. Even so, CBN still outperforms the variant with unfrozen BN in backbone by 1.8%.

**Instance segmentation and stronger backbones.** Results of object detection (Faster R-CNN (Ren et al., 2015)) and instance segmentation (Mask R-CNN (He et al., 2017)) with ResNet-50 and ResNet-101 are presented in Table 5. We can observe that our proposed CBN achieves performance comparable to syncBN and GN with R50 and R101 as the backbone on both Faster R-CNN and Mask R-CNN, which demonstrates that CBN is robust and versatile to various deep models and tasks.

## 4.3 ABLATION STUDY

**Effect of temporal window size $k$.** We conduct this ablation on ImageNet image classification and COCO object detection, with each GPU processing 4 images. Figure 3 presents the results. When $k = 1$, only the batch from the current iteration is utilized; therefore, CBN degenerates to the original BN. The accuracy suffers due to the noisy statistics on small batch sizes. As the window size $k$ gradually increases, more examples from recent iterations are utilized for statistics estimation, leading to greater accuracy. Accuracy saturates at $k = 8$ and even drops slightly. For more distant iterations, the network weights differ more substantially and the Taylor polynomial approximation becomes less accurate.

On the other hand, it is empirically observed that the original BN saturates at a batch size of 16 or 32 for numerous applications (Peng et al., 2018; Wu & He, 2018), indicating that the computed statistics become accurate. Thus, a temporal window size of $k = \min(\lceil \frac{16}{\text{bs per GPU}} \rceil, 8)$ is suggested.

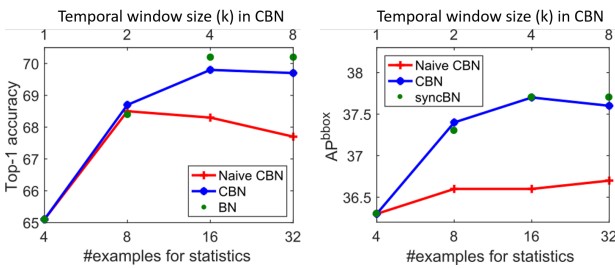

Figure 3: The effect of **temporal window size (k)** on ImageNet (ResNet-18) and COCO (Faster R-CNN with ResNet-50 and FPN) with #bs/GPU = 4 for CBN and Naive CBN. Naive CBN directly utilizes statistics from recent iterations, while BN uses the equivalent #examples as CBN for statistics computation.

**Effect of compensation.** To study this, we compare CBN with 1) a naive baseline where statistics from recent iterations are directly aggregated without compensation via Taylor polynomial, referred to as Naive CBN; and 2) the original BN applied with the same effective example number as CBN (i.e., its batch size per GPU is set to the product of the batch size per GPU and the temporal window size of CBN), which does not require any compensation and serves as an upper performance bound.

The experimental results are also presented in Figure 3. CBN clearly surpasses Naive CBN when the previous iterations are included. Actually, Naive CBN fails when the temporal window size grows to $k = 8$ as shown in Figure 3(a), demonstrating the necessity of compensating for changing network weights over iterations. Compared with the original BN upper bound, CBN achieves similar accuracy at the same effective example number. This result indicates that the compensation using a low-order Taylor polynomial by CBN is effective.

Figure 4 presents the train and test curves of CBN, Naive CBN, BN-bs4, and BN-bs16 on ImageNet, with 4 images per GPU and a temporal window size of 4 for CBN, Naive CBN, and BN-bs4, and 16 images per GPU for BN-bs16. The train curve of CBN is close to BN-bs4 at the beginning, and approaches BN-bs16 at the end. The reason is that we adopt a burn-in period to avoid the disadvantage of rapid statistics change at beginning of training. The gap between the train curve of Naive CBN and CBN shows that

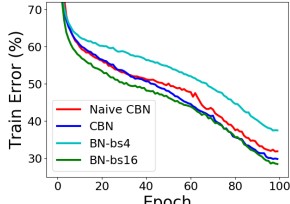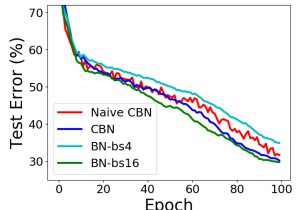

Figure 4: Training and test curves for CBN, Naive CBN, and BN on ImageNet, with batch size per GPU of 4 and temporal window size $k = 4$ for CBN, Naive CBN, and BN-bs4, and batch size per GPU of 16 for BN-bs16. Thus, the plot of BN-bs16 is the ideal bound.

Naive CBN cannot even reach a good convergence on the training set. The test curve of CBN is close to BN-bs16 at the end, while Naive CBN exhibits considerable jitter. All these phenomena indicate the effectiveness of our proposed Taylor compensation.

**Additional computational overhead and memory footprint.** As the inference stage of CBN is the same as BN, we only need to compare the computational overhead and memory footprint at the training time, shown in Table 6. The extra computational overhead mainly includes calculations of the statistics' respective gradients, Taylor compensations, and averaging operations. For the extra memory, the statistics ($\mu$ and $\nu$), their respective gradients, and the network parameters ($\theta_{t-1} \cdots \theta_{t-(k-1)}$) of previous iterations are all stored when applying CBN.

|  |  | ImageNet | | COCO | |
|---|---|---|---|---|---|
|  |  | BN | CBN | BN | CBN |
| GFLOPs | Base model | 5.96 | 5.96 | 5155.1 | 5809.7 |
|  | Taylor expansion | - | 0.21 | - | 654.6 |
|  | Total | 5.96 | 6.17 | 5155.1 | 5809.7 |
| Memory (GB) | Feature map | 0.15 | 0.45 | 14.1 | 15.1 |
|  | Net params | 0.09 | 0.21 | 0.3 | 0.6 |
|  | Total | 0.24 | 0.66 | 14.4 | 15.7 |

Table 6: Comparison of theoretical **FLOPs and memory footprint** between CBN and original BN in both forward and backward passes at training time.

From these results, the additional overhead of CBN is seen to be minor.

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

## A    ALGORITHM OUTLINE

Algorithm 1 presents an outline of our proposed Cross-Iteration Batch Normalization (CBN).

---

**Algorithm 1:** Cross-Iteration Batch Normalization(CBN)

---

**Input:** Feature responses of a network node of the $l$-th layer at the $t$-th iteration $\{x_{t,i}^l(\theta_t)\}_{i=1}^m$,
   network weights $\{\theta_{t-\tau}^l\}_{\tau=0}^{k-1}$, statistics $\{\mu_{t-\tau}^l(\theta_{t-\tau})\}_{\tau=1}^{k-1}$ and $\{v_{t-\tau}^l(\theta_{t-\tau})\}_{\tau=1}^{k-1}$, and gradients
   $\{\partial\mu_{t-\tau}(\theta_{t-\tau})/\partial\theta_{t-\tau}^l\}_{\tau=1}^{k-1}$ and $\{\partial v_{t-\tau}(\theta_{t-\tau})/\partial\theta_{t-\tau}^l\}_{\tau=1}^{k-1}$ from most recent $k-1$ iterations

**Output:** $\{y_{t,i}^l(\theta_t) = \text{CBN}(x_{t,i}^l(\theta_t))\}$

1  $\mu_t(\theta_t) \leftarrow \frac{1}{m}\sum_{i=1}^m x_{t,i}(\theta_t),\ v_t(\theta_t) \leftarrow \frac{1}{m}\sum_{i=1}^m x_{t,i}^2(\theta_t)$       //statistics on the current iteration

2  **for** $\tau \in \{1,\dots,k\}$ **do**

3       $\mu_{t-\tau}^l(\theta_t) \leftarrow \mu_{t-\tau}^l(\theta_{t-\tau}) + \frac{\partial\mu_{t-\tau}^l(\theta_{t-\tau})}{\partial\theta_{t-\tau}^l}(\theta_t^l - \theta_{t-\tau}^l)$       //approximation from recent iterations

4       $v_{t-\tau}^l(\theta_t) \leftarrow v_{t-\tau}^l(\theta_{t-\tau}) + \frac{\partial v_{t-\tau}^l(\theta_{t-\tau})}{\partial\theta_{t-\tau}^l}(\theta_t^l - \theta_{t-\tau}^l)$       //approximation from recent iterations

5  **end**

6  $\bar{\mu}_{t,k}^l(\theta_t) \leftarrow \frac{1}{k}\sum_{\tau=0}^{k-1}\mu_{t-\tau}^l(\theta_t)$       //averaging over recent iterations

7  $\bar{v}_{t,k}^l(\theta_t) \leftarrow \frac{1}{k}\sum_{\tau=0}^{k-1}\max\left[v_{t-\tau}^l(\theta_t),\mu_{t-\tau}^l(\theta_t)^2\right]$       //validation and averaging over recent iterations

8  $\bar{\sigma}_{t,k}^l(\theta_t)^2 \leftarrow \bar{v}_{t,k}^l(\theta_t) - \bar{\mu}_{t,k}^l(\theta_t)^2$

9  $\hat{x}_{t,i}^l(\theta_t) = \frac{x_{t,i}^l(\theta_t)-\bar{\mu}_{t,k}^l(\theta_t)}{\sqrt{\bar{\sigma}_{t,k}^l(\theta_t)^2+\varepsilon}}$       //normalize

10  $y_{t,i}^l(\theta_t) \leftarrow \gamma\hat{x}_{t,i}^l(\theta_t) + \beta$       //scale and shift

---

## B    EFFICIENT IMPLEMENTATION OF $\partial\mu_{t-\tau}^l(\theta_{t-\tau})/\partial\theta_{t-\tau}^l$ AND $\partial v_{t-\tau}^l(\theta_{t-\tau})/\partial\theta_{t-\tau}^l$

Let $C^l$ and $C^{l-1}$ denote the channel dimension of the $l$-th layer and the $(l-1)$-th layer, respectively, and $K$ denotes the kernel size of $\theta_{t-\tau}^l$. $\mu_{t-\tau}^l$ and $v_{t-\tau}^l$ are thus of $C^l$ dimensions in channels, and $\theta_{t-\tau}^l$ is a $C^l \times C^{l-1} \times K$ dimensional tensor. A naive implementation of $\partial\mu_{t-\tau}^l(\theta_{t-\tau})/\partial\theta_{t-\tau}^l$ and $\partial v_{t-\tau}^l(\theta_{t-\tau})/\partial\theta_{t-\tau}^l$ involves computational overhead of $O(C^l \times C^l \times C^{l-1} \times K)$. Here we find that the operations of $\mu$ and $v$ can be implemented efficiently in $O(C^{l-1} \times K)$ and $O(C^l \times C^{l-1} \times K)$, respectively, thanks to the averaging of feature responses in $\mu$ and $v$.

Here we derive the efficient implementation of $\partial\mu_{t-\tau}^l(\theta_{t-\tau})/\partial\theta_{t-\tau}^l$. That of $\partial v_{t-\tau}^l(\theta_{t-\tau})/\partial\theta_{t-\tau}^l$ is about the same. Let us first simplify the notations a bit. Let $\mu^l$ and $\theta^l$ denote $\mu_{t-\tau}^l(\theta_{t-\tau})$ and $\theta_{t-\tau}^l$ respectively, by removing the irrelevant notations for iterations. The element-wise computation in the forward pass can be computed as

$$\mu_j^l = \frac{1}{m}\sum_{i=1}^m x_{i,j}^l, \tag{13}$$

where $\mu_j^l$ denotes the $j$-th channel in $\mu^l$, and $x_{i,j}^l$ denotes the $j$-th channel in the $i$-th example. $x_{i,j}^l$ is computed as

$$x_{i,j}^l = \sum_{n=1}^{C^{l-1}}\sum_{k=1}^K \theta_{j,n,k}^l \cdot y_{i+\text{offset}(k),n}^{l-1}, \tag{14}$$

where $n$ and $k$ enumerate the input feature dimension and the convolution kernel index, respectively, offset$(k)$ denotes the spatial offset in applying the $k$-th kernel, and $y^{l-1}$ is the output of the $(l-1)$-th layer.

The element-wise calculation of $\partial \mu^l / \partial \theta^l \in \mathbb{R}^{C^l \times C^l \times C^{l-1} \times K}$ is as follows, taking Eq. (13) and Eq. (14) into consideration:

$$
\begin{aligned}
\left[\frac{\partial \mu^l}{\partial \theta^l}\right]_{j,q,p,\eta} &= \frac{\partial \mu_j^l}{\partial \theta_{q,p,\eta}^l} \\
&= \frac{\partial \frac{1}{m} \sum_{i=1}^m x_{i,j}^l}{\partial \theta_{q,p,\eta}^l} \\
&= \frac{\partial \frac{1}{m} \sum_{i=1}^m \sum_{n=1}^{C^{l-1}} \sum_{k=1}^K \theta_{j,n,k}^l \cdot y_{i+\text{offset}(k),n}^{l-1}}{\partial \theta_{q,p,\eta}^l} \\
&= \begin{cases} \frac{1}{m} \sum_{i=1}^m y_{i+\text{offset}(\eta),p}^{l-1} & , j = q \\ 0 & , j \neq q \end{cases} .
\end{aligned}
\tag{15}
$$

Thus, $[\frac{\partial \mu^l}{\partial \theta^l}]_{j,q,p,\eta}$ takes non-zero values only when $j = q$. This operation can be implemented efficiently in $O(C^{l-1} \times K)$. Similarly, the calculation of $\partial v^l / \partial \theta^l$ can be obtained in $O(C^l \times C^{l-1} \times K)$.

## C  ADDITIONAL EXPERIMENTS

**CIFAR-10** is selected for the experiments in this section. It consists of 50k training images and 10k test images from 10 classes. We train the standard ResNet-18 for 160 epochs on one GPU by SGD. The momentum and weight decay parameters are set to 0.9 and 0.0001, respectively. We experiment with batch sizes of 32, 16, 8, 4, and 2 images per GPU. The learning rate is scaled linearly to different batch sizes, following the practice in (Peng et al., 2018). The initial learning rate is $0.025 * N/32$ for a batch size per

| Trials | 1 | 2 | 3 | 4 | 5 | overall |
|---|---|---|---|---|---|---|
| BN-bs16 | 95.3 | 95.0 | 95.4 | 95.3 | 95.3 | 95.26±0.14 |
| BN-bs4 | 93.6 | 93.5 | 93.6 | 93.6 | 93.8 | 93.62±0.11 |
| BRN | 94.4 | 94.7 | 94.5 | 94.3 | 94.4 | 94.46±0.15 |
| GN | 94.3 | 94.0 | 94.4 | 94.2 | 94.4 | 94.26±0.17 |
| CBN | 95.0 | 95.0 | 94.6 | 95.0 | 95.2 | 94.96±0.22 |

Table 7: Top-1 accuracy of ResNet-18 with **different trials** on CIFAR-10. The batch size per GPU is 16 and 4 for BN-bs16 and the other methods, respectively.

iteration of $N$. The learning rate is divided by 10 at epochs 80 and 120. The images are of $32 \times 32$ pixels with per-image standardization in both training and inference. Random flipping is applied in training.

We report the results of BN, BRN, GN, CBN with five trials on CIFAR-10, as shown in Table 7. CBN has the smallest gap with BN-bs16 compared to BN-bs4, BRN, and GN. This result is consistent with previous experiments on ImageNet and COCO. Also the *std* is tiny, indicating that performance on CIFAR-10 is stable enough for some empirical studies.

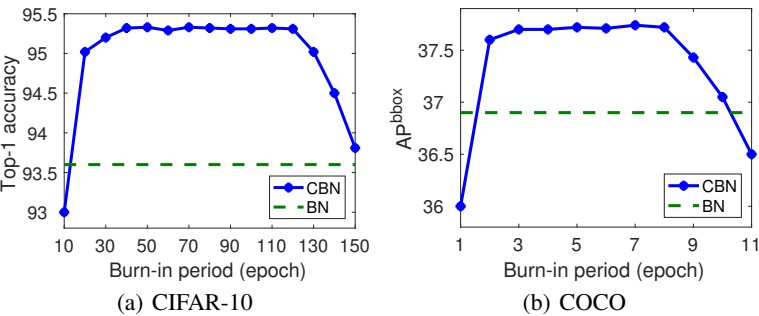

(a) CIFAR-10  (b) COCO

Figure 5: Results of **different burn-in periods (in epochs)** on CBN, with batch size per iteration of 4, on CIFAR-10 and COCO.

**On the burn-in period length $T_{\text{burn-in}}$.** We further study the influence of varying the burn-in period length $T_{\text{burn-in}}$, at 4 images per GPU on both CIFAR-10 image classification (ResNet-18) and COCO object detection (Faster R-CNN with FPN and ResNet-50).

Figure 5(a) and 5(b) present the results. When the burn-in period is too short, the accuracy suffers. This is because at the beginning of training, the network weights change rapidly, causing the

compensation across iterations to be less effective. On the other hand, the accuracy is stable for a wide range of burn-in periods $T_{\text{burn-in}}$ that are not too short.

**On the effect of using more than one layer.** The efficient implementation is no longer applicable when more than one layer of compensation is adopted. Therefore, we only conduct a two-layer experiment of ResNet-18 on CIFAR-10 in consideration of the heavy extra memory and computational overhead. CBN using two layers for compensation achieves 95.0 on CIFAR-10 (batch size=4, k=4), which is comparable to CBN using only one layer. As using more layers does not further improve performance but consumes more FLOPs, we adopt one-layer compensation on CBN in practice.

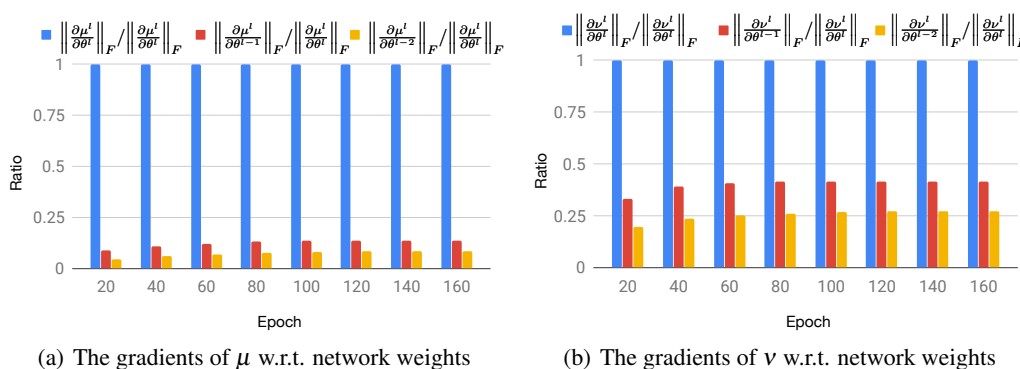

(a) The gradients of $\mu$ w.r.t. network weights     (b) The gradients of $\nu$ w.r.t. network weights

Figure 6: Comparison of gradients of statistics w.r.t. current layer vs. that w.r.t. previous layers on CIFAR-10.

**On the gradients from different layers.** The key assumption in Eq. (7) and Eq. (8) is that for a node at the $l$-th layer, the gradient of its statistics with respect to the network weights at the $l$-th layer is larger than that of weights from the prior layers, i.e., $||\frac{\partial \mu_{t-\tau}^l(\theta_{t-\tau})}{\partial \theta_{t-\tau}^l}||_F \gg ||\frac{\partial \mu_{t-\tau}^l(\theta_{t-\tau})}{\partial \theta_{t-\tau}^r}||_F$ and $||\frac{\partial \nu_{t-\tau}^l(\theta_{t-\tau})}{\partial \theta_{t-\tau}^l}||_F \gg ||\frac{\partial \nu_{t-\tau}^l(\theta_{t-\tau})}{\partial \theta_{t-\tau}^r}||_F$ for $r < l$, where $|| \cdot ||_F$ denotes the Frobenius norm. Here we examine this assumption empirically for networks trained on CIFAR-10 image recognition.

Figure 6 presents the computed ratio of $||\frac{\partial \mu_{t-\tau}^l(\theta_{t-\tau})}{\partial \theta_{t-\tau}^l}||_F / ||\frac{\partial \mu_{t-\tau}^l(\theta_{t-\tau})}{\partial \theta_{t-\tau}^r}||_F$ and $||\frac{\partial \nu_{t-\tau}^l(\theta_{t-\tau})}{\partial \theta_{t-\tau}^l}||_F / ||\frac{\partial \nu_{t-\tau}^l(\theta_{t-\tau})}{\partial \theta_{t-\tau}^r}||_F$ for $r \le l$, at different training epochs. The results suggest that $||\frac{\partial \mu_{t-\tau}^l(\theta_{t-\tau})}{\partial \theta_{t-\tau}^l}||_F \gg ||\frac{\partial \mu_{t-\tau}^l(\theta_{t-\tau})}{\partial \theta_{t-\tau}^r}||_F$ and $||\frac{\partial \nu_{t-\tau}^l(\theta_{t-\tau})}{\partial \theta_{t-\tau}^l}||_F \gg ||\frac{\partial \nu_{t-\tau}^l(\theta_{t-\tau})}{\partial \theta_{t-\tau}^r}||_F$ hold for $r < l$, thus validating the approximation in Eq. (7) and Eq. (8).

We also study the gradients of non-ResNet models. $||\frac{\partial \mu_{t-\tau}^l(\theta_{t-\tau})}{\partial \theta_{t-\tau}^{l-2}}||_F / ||\frac{\partial \mu_{t-\tau}^l(\theta_{t-\tau})}{\partial \theta_{t-\tau}^l}||_F$ and $||\frac{\partial \nu_{t-\tau}^l(\theta_{t-\tau})}{\partial \theta_{t-\tau}^{l-2}}||_F / ||\frac{\partial \nu_{t-\tau}^l(\theta_{t-\tau})}{\partial \theta_{t-\tau}^l}||_F$ on VGG-16 and InceptionV3 are (0.22 and 0.46) and (0.17 and 0.38), respectively, which is similar to ResNet-18 (0.13 and 0.40), indicating that the assumption should also hold for VGG and the Inception series.

