# OpenReview forum: "Cross-Iteration Batch Normalization"
_ICLR.cc/2020/Conference — Reject_

### Official Review · AnonReviewer1 · 2019-10-21
**Official Blind Review #1**

**Rating:** 3

**Review:**

Paper summary: This paper proposes a new normalization technique specially designed for settings with small mini-batch sizes (where previous methods like BatchNorm are known to suffer). The approach aggregates mean/variance statistics from previous iterations, weighted based on the Taylor expansion, to get a better estimate of population statistics. The authors evaluate their approach on ImageNet classification, and object detection and instance segmentation on the COCO dataset.



The paper is clearly written and explores an interesting idea---aggregating mini-batch statistics across iterations. That being said, I am not convinced by the utility of the proposed approach since it doesn’t offer over significant benefits over prior approaches designed for the small mini-batch setting (e.g., Group Normalization)---either in terms of empirical performance, implementation complexity, or lower computational/memory requirements.

Specific comments/questions:

1. Why do the authors not include the Kalman Normalization baseline in the paper? Based on my understanding, it is also designed for the low-sample regime (and the original paper also conducts experiments on ImageNet/COCO). Also the BRN baseline is included in Table 3 for ImageNet but is missing from the COCO experiments. It is important to thoroughly compare to other normalization techniques specifically designed for this regime to clearly highlight relative benefits of the proposed approach.

2. The authors mention that they average performance across 5 trials but omit confidence intervals in their Figures/Tables. Since the difference in performance between the different approaches compared is small, I think confidence intervals should be reported to see whether improvements are within statistical error margins. In fact, for CIFAR-10 based on Table 7 in the Appendix, performance of batch renormalization, group normalization and CBN is essentially the same.

3. In Table 3, how many iterations was the BRN aggregation performed over? Was it also chosen to ensure effective number of samples was not less than 16 (like CBN). What do Figures 3 and 4 look like for BRN?

4. One thing I find interesting (the authors do not discuss this specifically) about Figure 5 in the Appendix is that both for CIFAR and COCO (the two datasets for which plots are provided), CBN helps as long as it is used before the final learning rate drop. Specifically, choosing a burn-in period as large as 120 for CIFAR and 8 for COCO is fine (in fact, probably the best) as long as you turn on CBN before the final learning rate drop. This makes me curious whether BN in the low-sample regime only suffers in terms of generalization performance in the final stages of training (compared to low-sample alternatives like GN/CBN, etc).



The authors should include other recent benchmarks (Kalman normalization and BRN in the omitted Tables) and error bars to make the change in performance clearer. It would also be interesting to see whether (and probably make the paper stronger if) alternatives like GN/KN benefit from being combined with the proposed scheme to aggregate statistics over time.

Overall, while the proposed approach is novel, its performance is comparable to prior approaches, with the disadvantage of an additional computational/memory footprint. Thus, I am not yet convinced about how useful/interesting this approach would be to the community.


**Experience Assessment:**

I have published one or two papers in this area.

**Review Assessment: Checking Correctness Of Derivations And Theory:**

I assessed the sensibility of the derivations and theory.

**Review Assessment: Checking Correctness Of Experiments:**

I carefully checked the experiments.

**Review Assessment: Thoroughness In Paper Reading:**

I read the paper thoroughly.

---

> ### Author Response · Authors · 2019-11-14
> **Response to AnonReviewer1**
>
> We thank the reviewer for the careful reviews and constructive suggestions. We feel we can well address the concerns of R#1, and hope R#1 give a second thought about the paper.
>
> 1. We present our experimental results with BRN as follows:
> |         Method         | ImageNet-Top1(%) | COCO-mAP_bbox |
> |-------------------------|---------------------------|-------------------------|
> | BN-bs16/syncBN |               70.2             |             37.7             |
> | BN-bs1                  |               65.1             |             36.3             |
> | BRN                       |               67.9             |             37.5             |
> | GN                         |               69.0             |             37.8             |
> | CBN                       |               69.8             |             37.7             |
> For KN, we tried our best to re-implement it and also asked the author for help, but failed to reproduce the result without a response from the author.
>
> 2. As mentioned in the paper, the variance between different trials is negligible for both ImageNet and COCO. For CIFAR-10, we show the error bar in Table 7 of the Appendix.
>
> 3. For BRN, the statistics are updated in a "moving" policy. So there is no parameter similar to “aggregation iteration”.
>
> 4. Many thanks to the reviewer for sharing this interesting perspective. To perform this further exploration, we design a new ablation experiment to remove other influences: we first train the model on COCO with standard BN and a small batch size, then switch BN to syncBN (the learning rate will be divided by 10 at epoch-9 and epoch-11). We present the experimental results as follows:
> | Resume Epoch | Epoch-8  | Epoch-9  | Epoch-10 | Epoch-11 |
> |----------------------|-------------|-------------|---------------|--------------|
> | mAP_bbox         |     37.7    |     37.7     |      37.6      |      37.3     |
>
> Interestingly, as mentioned by the reviewer, BN in the low-sample regime only hurts the generalization performance in the final stage after the learning rate drops. We leave this as a direction for future study.
>
> The major concern on “how useful/interesting this approach would be to the community” is addressed in the comment to all reviewers, please check it. Thanks!
>
> We hope that your concerns are addressed.

---

> > ### Comment · AnonReviewer1 · 2019-11-15
> > **Re: Author Response**
> >
> > Thank you for addressing my questions. During the rebuttal the authors did report confidence intervals for an experiment on CIFAR-10 in the appendix, but not for their main ImageNet/COCO experiments in the paper. As I mentioned in my review earlier, given that the differences between the various approaches are small, reporting variance over repeated runs is critical to comparing them. Thus, I still have my reservations regarding the proposed approach---(1) improvement over SOTA is small and (2) comes with additional overheads. This limits the utility of the approach in my opinion.

---

> > > ### Author Response · Authors · 2019-11-15
> > > **Response to AnonReviewer1**
> > >
> > > Thanks a lot for your response. After reading our response, we hope R#1 give a second thought about the paper.
> > >
> > > 1. As the performance variance on COCO and ImageNet is so low, most of the previous works, such as previous normalization methods [1,2,3], image recognition methods on ImageNet [4,5,6], object detection methods on COCO [7,8,9], all ignore the error bar on COCO and ImageNet. So is the variance of our proposed CBN.
> > >
> > > 2. In our perspective, we propose a general idea orthogonal to all previous normalization methods, allowing all methods to borrow and compensate the statistics across iterations. This also provides a new perspective that the small batch size issue could be solved in this way. In our mind, this is valuable to the community, and has opportunities to inspire other researchers to dig deeper and learn from our paper.
> > >
> > > 3. In this paper, we focus on addressing the small batch size issue on BN, so BN with large batch size is the upper bound of our paper. In the most widely used setting (bs=4 on COCO and ImageNet), our proposed CBN could achieve the upper bound performance. Even in the extreme case, bs=1, the performance drop of CBN is acceptable. Hence in our perspective, the performance is not the drawback of our proposed CBN, because it is already close to the upper bound.
> > >
> > > 4. Although different normalization methods (GN, CBN, SyncBN) achieve on par performance on COCO, they all have specific additional overheads or other disadvantages. Compared to BN, GN is 2 times slower for inference (reference to our general response to all reviewers); SyncBN highly relies on the low-latency and high-bandwidth communication, which can only be addressed with hardware. Our method has only 3.5% and 12.6% computational overheads for training on ImageNet and COCO, but no overheads for inference, which is totally acceptable in our mind.
> > >
> > >
> > > [1] MegDet: A Large Mini-Batch Object Detector, CVPR 2018
> > > [2] Group Normalization, ECCV 2018
> > > [3] Kalman Normalization: Normalizing Internal Representations Across Network Layers, NeurIPS 2018
> > > [4] Deep Residual Learning for Image Recognition, CVPR 2016
> > > [5] Densely Connected Convolutional Networks, CVPR 2017
> > > [6] Squeeze-and-Excitation Networks, CVPR 2018
> > > [7] Faster R-CNN: Towards Real-Time Object Detection with Region Proposal Networks, NIPS 2015
> > > [8] Mask R-CNN, ICCV 2017
> > > [9] Rethinking ImageNet Pre-training, ICCV 2019

---

### Official Review · AnonReviewer3 · 2019-10-22
**Official Blind Review #3**

**Rating:** 6

**Review:**

The paper tackles the problem of batch normalization (BN) instability when using small batch sizes. As the fix, authors propose the following modifications to BN:
 1) Use previous iterations statistics to virtually increase batch size
 2) Correction via Taylor series linearization of of the previous iterations to compensate weight changes


Paper shows in experiments on classification (ImageNet), detection (COCO) and segmentation (COCO), that the proposed method works on par with other normalization methods. Paper also shows that the proposed method does not work well for the beginning of the training and use "burn-in" period when standard BN is used instead of proposed CBN.


Questions:

 1) Is it possible to use proposed method with batch size = 1? This could be a killer feature.
 2) Why the CBN/BN memory footprints ratios for ImageNet and COCO (Table 6) are so different? (2.75 x for ImageNet vs 1.09 for COCO).
 It seems that for ImageNet there is no benefit of using CBN, because it takes 3x more memory than BN (0.66 vs 0.24 Gb), so one can just increase batch size 3 times for standard BN.
 3) Paper reports results for "validation set", yet "hyper-parameters were set by cross-validation". Could you please specify, how was cross-validation done?
 4) There is a practice of gradient accumulation:  doing multiple forward-backward passes, then apply optimization step once. Could you please comment, how such practice may possible interfere with proposed weight compensation?
 Wouldn`t it be benefitial to do similar practice for cross-iteration BN, so that no weight compensation be needed for such case?
5) What are potential use cases of CBN compared to use  BN for big batches and GN for small batches, even bs = 1?


Overall I think that paper is OK, but don`'t see practical applications where it is beneficial to use CBN instead of BN (for big batches) or GN (for small batches, even bs = 1).  I may be willing to increase my score, if my questions would be addressed.


####

My concerns were addressed in the rebuttal and I am happy to increase my rating to weak accept.


**Experience Assessment:**

I have read many papers in this area.

**Review Assessment: Checking Correctness Of Derivations And Theory:**

I assessed the sensibility of the derivations and theory.

**Review Assessment: Checking Correctness Of Experiments:**

I assessed the sensibility of the experiments.

**Review Assessment: Thoroughness In Paper Reading:**

I read the paper thoroughly.

---

> ### Author Response · Authors · 2019-11-14
> **Response to AnonReviewer3**
>
> We thank the reviewer for the constructive suggestions and feedback. We feel we can well address the concerns of R#3, and hope R#3 give a second thought about the paper.
>
> Due to the limited time for rebuttal, we present the experiments of bs=1 on CIFAR-10 as follows (with the same settings mentioned in our paper's Appendix):
> | Method | Acc(%) |
> |------------|-----------|
> | BN-bs16|   95.3   |
> | BN-bs1 |   60.0* |
> | BRN      |   83.1* |
> | GN        |   93.4   |
> | CBN      |   93.3   |
> * denotes may not converge.
>
> 1. From the table above, our proposed CBN could work well and achieve performance comparable to GN when bs=1.
>
> 2. That's because the overall architecture of Mask-RCNN is quite different from the original ResNet, which makes it not appropriate to directly compare the CBN/BN ratio between ImageNet and COCO. If we compare the absolute increase of memory footprint 0.42(=0.66-0.24) and 1.3(=15.7-14.4), we would find the ratio is about 3.5x and this was close to the ratio of different input sizes (400*672)/(224*224).
> Object detection on COCO is a real application scenario that suffers from the small batch size issue, costing a really large memory footprint with a vanilla network architecture. In this case, the relative increase of CBN on memory footprint, 9%, is really small.
>
> 3. Sorry for the typo, we will correct this in the revision.
>
> 4. The gradients can be accumulated in multiple forward and backward passes in a sequential manner, but the statistics cannot. If the BN layer in one pass wants the statistics from the data in other forward passes, it would wait for other passes to send back their statistics during their forward passes. So all the forward passes need to have the exact same pace to guarantee that their statistics are synced in each layer. This results in SyncBN (multiple parallel forward backward passes), an alternate choice for the original BN. The main drawbacks of SyncBN are already mentioned in our paper, like the heavy requirements of low-latency and high-bandwidth communication between multiple GPUs.
> .
> 5. The major concern on “where it is beneficial to use CBN instead of GN (for small batches, even bs = 1)” is addressed in the comment to all reviewers, please check it. Thanks!
>
> We hope that your concerns are addressed.

---

> > ### Comment · AnonReviewer3 · 2019-11-15
> > **Thank you for additional experiments!**
> >
> > I find experiments with bs=1 interesting.  Also thanks for the pointing out slower GN inference compared to the BN/CBN, it is not obvious and worth adding to the paper.
> >
> > >3. Sorry for the typo, we will correct this in the revision.
> > So how were hyperparameters set?

---

> > > ### Author Response · Authors · 2019-11-15
> > > **Response to AnonReviewer3**
> > >
> > > It is inspiring that our response reduces most of your concerns. Thanks again for your constructive suggestions that would improve our submission a lot.
> > >
> > > It is common practice to tune the hyper-parameter on the validation set of COCO and ImageNet.
> > >
> > > For our approach, we only have two extra hyper-parameters, the window size k and the burn-in period T.
> > >
> > > For the window size k, we study the effectiveness of k in Section 4.3. Based on the results, we simply follow min(ceil(16/bs per GPU), 16) to set k.
> > >
> > > For the burn-in period T, the ablation experiment is shown in Appendix.C. The results indicate that the performance are quite stable if T is set between the epoch after warmup and the epoch when the first learning rate drop.
> > >
> > > For other hyper-parameters like momentum and eps, our CBN is absolutely the same with BN.

---

### Official Review · AnonReviewer2 · 2019-10-23
**Official Blind Review #2**

**Rating:** 6

**Review:**

This paper proposes a novel Cross-Iteration Batch Normalization (CBN) to address the limitation of BN in the case of small mini-batch sizes. Different from existing methods, CBN exploits the statistics cross different iterations to obtain more accurate estimates of the data statistics. Specifically, the proposed CBN uses Taylor polynomials to approximate the statistics using the information of multiple recent iterations. The experiments on both image classification and object detection tasks demonstrate the effectiveness of the proposed method.

Please see my detailed comments below.

Positive points:

1. The authors propose a novel method that exploits the information from different iterations to estimate the normalization statistics more accurately.

2. Unlike existing methods, the authors use Taylor polynomials to cast the information of previous iterations into the current iteration. The proposed method is theoretically sound.

3. The experiments on both image classification and object detection tasks show the superiority of the proposed CBN over the considered baseline methods.

Negative points:

1. The authors only conduct experiments on a single image classification model ResNet-18. More deep architectures should be considered in the experiments, e.g., DenseNet, MobileNet, etc.

2. Several state-of-the-art normalization methods should be compared in the experiment of " sensitivity to batch size", including SN [1] and DN [2].

3. One limitation of the proposed CBN method is that it takes much more computational complexity and memory consumption than the baseline BN method (See Table 6). It is not clear whether the performance improvement comes from the increased computational cost.

4. It would be stronger to compare the inference time of different normalization methods.

5. This paper considers the case of small mini-batch sizes. However, the authors only investigate 5 different batch sizes, such as (32, 16, 8, 4, 2). What would happen if the authors set the batch size to 1?

6. The differences from a closely related work [3] should be discussed. This paper exploits the similar idea of computing the statistics of the current iteration by exploiting the statistics of multiple recent iterations.

7. Some typos

(1) In the experiment of Comparison of feature normalization methods, “Similar as the results ...” should be “Similar to the results ...”

(2) In the section of Effect of compensation, “The train curve ... “ should be “The training curve ...”

Some References

[1] "Differentiable Learning-to-normalize via Switchable Normalization." ICLR, 2019.
[2] "Differentiable Dynamic Normalization for Learning Deep Representation." ICML, 2019.
[3] "Double Forward Propagation for Memorized Batch Normalization." AAAI, 2018.



**Experience Assessment:**

I have published one or two papers in this area.

**Review Assessment: Checking Correctness Of Derivations And Theory:**

I carefully checked the derivations and theory.

**Review Assessment: Checking Correctness Of Experiments:**

I carefully checked the experiments.

**Review Assessment: Thoroughness In Paper Reading:**

I read the paper thoroughly.

---

> ### Author Response · Authors · 2019-11-14
> **Response to AnonReviewer2**
>
> We thank the reviewer for the careful reviews and constructive suggestions. We address the questions as follows.
>
> 1. Any BN-equipped deep CNN can enjoy the benefits brought by our CBN in the case of small mini-batch sizes. Following the reviewer's suggestion, we conducted additional experiments on other deep architectures, including DenseNet and MobileNet.
> | Method | DenseNet121 | MobileNet |
> |------------|--------------------|----------------|
> | BN-bs16|        95.5          |      88.6       |
> | BN-bs4  |        94.8          |      86.4       |
> | CBN       |        95.4          |      88.2       |
> Due to limited time, we conducted the experiment on CIFAR-10 with the same settings mentioned in our paper's Appendix.
>
> 2. To the best of our knowledge, both SN and DN explore the combination of existing normalization approaches, but do not focus on the problem about the reduced effectiveness of batch normalization in the case of small mini-batch sizes. So SN and DN are not natural baselines for our CBN.
>
> 3. The relative computation cost introduced by CBN compared with an entire deep neural network is very small. To be specific, the relative increase of FLOPS from CBN compared to BN on ImageNet and COCO is 3.5% and 12.6%, respectively. Besides, the increase in computation cost from CBN is to obtain more accurate statistics, which does not increase the capacity of the deep networks. Hence, the performance improvement of CBN is not from the additional computational cost.
>
> 4. As both CBN and BN use running mean/variance in the inference stage, the inference time of CBN is strictly equal to that of BN.
>
> 5. Thanks for your suggestion. Because of the limited time for rebuttal, we present the experiments of bs=1 on CIFAR-10 as follows (with the same settings mentioned in our paper's Appendix):
> | Method | Acc(%) |
> |------------|-----------|
> | BN-bs16|   95.3   |
> | BN-bs1 |   60.0*  |
> | BRN      |   83.1*  |
> | GN        |   93.4    |
> | CBN      |   93.3    |
> * denotes may not converge.
> As shown in this table, our CBN still works well (on par with GN) when the batch size is extremely small (bs=1).
>
> 6. MBN first obtains the statistics from previous iterations similar to BRN or CBN-naive, and tries to obtain more accurate statistics by double forwarding the network. As the main cost of training a neural network is from forward and backward propagation, double forwarding the network significantly increases the computational cost compared to other normalization methods as well as ours.
> Our proposed cross-iteration idea is much more effective for obtaining accurate statistics. We not only borrow but also compensate for the statistics from previous iterations, and do not need the extra forwarding on the whole network.
>
> 7. Thanks for pointing out the typos. We will fix them in the next version.

---

### Public Comment · ~Ruosi_Wan2 · 2019-10-09
**Do you prepare to release code?**

Hi, I think your work is really fascinating, but I found it a little hard to reproduce your method. So I wonder if you prepare to release the code in a recent time? Thanks!

---

> ### Author Response · Authors · 2019-10-10
> **We will release the code in about 1-2 weeks**
>
> Hi Ruosi,
>
> Thanks a lot for your attention. Yes, open source is in our plan. Now we are working on cleaning up the codebase, and will release the code in about 1-2 weeks. According to the Algorithm 1 in Appendix A, it would be easy to reproduce our method CBN. If you have any questions about reproducing CBN, you can ask us directly.
>
> Best,
> Authors of CBN

---

> > ### Public Comment · ~Ruosi_Wan2 · 2019-10-12
> > **Problems when I reproduce CBN**
> >
> > Thanks for your response！
> >
> > When I try to reproduce CBN, the there are mainly two problem I met:
> >
> > 1.  Computing d\mu / d\theta, d\v / d\theta requires quantity of extra computation. In fact in my experiments, resnet18 with CBN is 4 times slower than vanilla resent18. I implemented the experiments using pytorch, and tried my best to follow the experiment settings mentioned in your paper, so I don't know if the reduction of training speed is normal.
> >
> > 2. The second problem is much more serious, my implementation of CBN doesn't converge. After 600,000 iterations (7.5 epoch, 4 samples per GPU), the top-1 error is still 98.9, while in vanilla BN (32 samples per GPU), after 7.5 epoch the top-1 error is 53.1. Of course it might caused by my wrong implementation of CBN, hence to eliminate the effect of incorrect of implementation of CBN, I did the following equivalence experiments:
> >
> > According to the paper, I think the core idea of CBN is to use a more accurate batch statistics \mu, \sigma at the current iteration during forward and backward propagation.  So in my equivalence experiment, the batch size is still 32 per GPU, but I divided the batch into 8 groups on each gpu, during forward, the input of BN layers still use \mu, \sigma computing over 32 samples on each GPU; during backward, the gradients only propagate within each group on each gpu. I think the equivalence experiment use even more accurate batch statistics than CBN, so its performance should not be worse than CBN. But in my experiment the training still do not converge.
> >
> > Here's a possible misunderstanding, you said  "In training, the loss gradients are backpropagated to the network weights and activations at the current iteration, i.e., θ_t^l and x_{t,i}^l(θ_t)." So should  θ_t^l  in equation 7, 8 be involved in backward propagation? For now I don't set equation 7, 8 in backward propagation for it's too slow.
> >
> > Now I'm still working on debugging and looking forward your further response to my confusion.
> >
> > Best,
> > Ruosi

---

> > > ### Author Response · Authors · 2019-10-14
> > > **Response**
> > >
> > > Hi Ruosi,
> > >
> > > A1. As we mentioned in the paper (last paragraph in Section 3.2), the naïve implementation of $\partial\mu/\partial\theta$ and $\partial\nu/\partial\theta$ could lead to high computational overhead $O(C^l\times C^l\times C^{l-1}\times K)$. Therefore, we provide two ways to implement this operation efficiently.
> > >
> > > 1.1 By utilizing the averaging over feature responses of $\mu$ and $\nu$, the calculations of $\partial\mu/\partial\theta$ and $\partial\nu/\partial\theta$ can be done in $O(C^{l-1}\times K)$ and $O(C^l\times C^{l-1}\times K)$, respectively, which is discussed in the Appendix B.
> > >
> > > 1.2 By reusing $\partial x/\partial\theta$ (where x is the input of CBN layer), the computational complexity of $\partial\mu/\partial\theta$ and $\partial\nu/\partial\theta$ can both be reduced to $O(C^l\times C^l)$. In other words, the convolution layer before CBN layer will do the math of $\partial x/\partial\theta$ while back-propagation. So $\partial\mu/\partial\theta$ and $\partial\nu/\partial\theta$ can be obtained by only calculating $\partial\mu/\partial x$ and $\partial\nu/\partial x$.
> > >
> > > A2. Do you use burn-in in your experiment? It’s important to apply CBN when the network weights are relatively stable.
> > >
> > > A3. The "In training…" sentence means that the gradients are only applied on the network weights and activations of current iteration. Since the weights of previous iterations are already detached from the network, there is no need to back-propagate or update them.
> > >
> > > Best,
> > > Authors of CBN

---

> > > > ### Public Comment · ~Ruosi_Wan2 · 2019-10-15
> > > > **Thanks for your advice**
> > > >
> > > > Very thanks for your response!
> > > >
> > > > I think I'd better wait for your public code.
> > > >
> > > > If it's truly easy to apply CBN, I think it's really a great work, which can not only provides a simple and effective way to deal with small batch issues, but also inspire us a lot on exploring the true effect of BN in training deep neural network.
> > > >
> > > > Best,
> > > > Ruosi

---

> > > > > ### Author Response · Authors · 2019-10-15
> > > > > **Thank you for your interest**
> > > > >
> > > > > Thank you for your interest. We will release the code as soon as possible.

---

### Author Response · Authors · 2019-10-26
**Code Release Anonymously**

Hi,

We released the code of CBN anonymously on Github ( https://github.com/CBN-code-release/mmdetection ).

Thanks,
Authors of CBN

---

### Author Response · Authors · 2019-11-14
**Response to all reviewers**

Thank all three reviewers for your constructive comments and feedback. It is exciting for us that the reviewers recognize the novelty of our approach, and we think this cross-iteration idea is new to the community and provides a new perspective that this small batch size issue could be solved in this way. As all the reviewers are curious about the potential use cases or advantages compared to other normalization methods, like GN, we address this concern as a whole.

1. The key novelty of CBN is its ability to obtain the statistics across iterations. Hence, most of the normalization approaches are orthogonal to our cross-iteration idea and can enjoy the benefits brought by cross-iteration statistics. For instance, we present the experiments of cross-iteration group normalization (CGN) with a shared channel size 4 for each group, shown in the table below. CGN can achieve slightly better performance than GN.
Due to the limited time for rebuttal, we present the experiments of bs=1 on CIFAR-10 as follows (with the same settings mentioned in our paper's Appendix):
| Method | Acc(%) |
|------------|-----------|
| BN-bs16|  95.3    |
| BN-bs1 |  60.0*   |
| GN        |  93.4     |
| CBN      |  93.3     |
| CGN      |  93.5     |

2. We agree that the proposed CBN would slightly increase the FLOPs (12.6% on COCO) and memory footprints (9% on COCO) in the training stage. But in the inference stage, our proposed CBN is strictly the same as BN with no increase of FLOPs, memory and inference time. So our CBN can benefit from the advantages of BN in the inference stage. For example, GN, which needs to compute the run time statistics in the inference stage, is significantly slower (1.47 times) than BN in the inference stage. These results are from Detectron [1], where the inference time (s/im) of BN and GN using Mask-RCNN-R50-FPN on COCO is 0.099 and 0.146, respectively.
Furthermore, previous work like [2] has shown another method called merged Conv-BN to merge the BN layer to the preceding convolution layer at the inference stage, with no harm to the performance but significantly speeding up the inference time. However, as GN needs to compute running statistics at the inference stage, it cannot benefit from this method. With this method, ResNet-50-BN is 2x faster than ResNet-50-GN with a standard 224*224 input size.

[1] Detectron: https://github.com/facebookresearch/Detectron/blob/master/projects/GN/README.md
[2] DSSD: Deconvolutional Single Shot Detector

---

### Decision · Program_Chairs · 2019-12-19

**Decision:**

Reject

**Comment:**

This paper proposes cross-iteration batch normalization, which is a strategy for maintaining statistics across iterations to improve the applicability of batch normalization on small batches of data.

The reviewers pointed out some strong points but also some weak points about the paper. The paper was judged to be novel and theoretically sound, and the paper was judged to be well-written.

However, there were some doubts regarding the relevance and significance of the work. Reviewers commented on being unconvinced by the utility of the approach, it being unclear when the proposed method is beneficial, and the relative small magnitude of the empirical improvement.

On the balance, the paper seems decent but not completely convincing. This means that with the current high competitiveness and selectivity of ICLR I unfortunately cannot recommend the manuscript for acceptance.